# ARTWHISPERER: A DATASET FOR CHARACTERIZING HUMAN-AI INTERACTIONS IN ARTISTIC CREATIONS

## ABSTRACT

As generative AI becomes more prevalent, it is important to study how human users interact with such models. In this work, we investigate how people use text-to-image models to generate desired target images. To study this interaction, we created ArtWhisperer, an online game where users are given a target image and are tasked with iteratively finding a prompt that creates a similar-looking image as the target. Through this game, we recorded over 50,000 human-AI interactions; each interaction corresponds to one text prompt created by a user and the corresponding generated image. The majority of these are repeated interactions where a user iterates to find the best prompt for their target image, making this a unique sequential dataset for studying human-AI collaborations. In an initial analysis of this dataset, we identify several characteristics of prompt interactions and user strategies. People submit diverse prompts and are able to discover a variety of text descriptions that generate similar images. Interestingly, prompt diversity does not decrease as users find better prompts. We further propose a new metric to quantify the *steerability* of AI using our dataset. We define steerability as the expected number of interactions required to adequately complete a task. We estimate this value by fitting a Markov chain for each target task and calculating the expected time to reach an adequate score in the Markov chain. We quantify and compare AI steerability across different types of target images and two different models, finding that images of cities and natural world images are more steerable than artistic and fantasy images. These findings provide insights into human-AI interaction behavior, present a concrete method of assessing AI steerability, and demonstrate the general utility of the ArtWhisperer dataset.

## 1 INTRODUCTION

Direct human interaction with AI models has become widespread following a number of technical innovations improving the quality of text-to-text Brown et al. (2020); Ouyang et al. (2022); Anil et al. (2023) and text-to-image models Rombach et al. (2022a); Ramesh et al. (2022), enabling the public release of high-quality AI-based services like ChatGPT chatGPT, Bard Bard, and Midjourney Midjourney. These models have seen rapid interest and adoption largely due to the ability of the general public to interact with and steer the AI in diverse contexts including engineering, creative writing, art, education, medicine, and law Dakhel et al. (2023); Nguyen & Nadi (2022); Ippolito et al. (2022); Cetinic & She (2022); Qadir (2023); Cascella et al. (2023); Sloan (2023).

A key challenge in developing these models is aligning their output to human inputs. This is made challenging by the broad domain of use cases as well as the diverse prompting styles of different users. Many approaches can be categorized as "prompt engineering," where specific strategies for prompting are used to steer a model Oppenlaender (2022); Liu & Chilton (2022); Zhou et al. (2022b); Wei et al. (2022); White et al. (2023). Great success has also been found by fine-tuning models using relatively small datasets to follow human instructions Ouyang et al. (2022), respond in a specific style Hu et al. (2021), or behave differently to specified prompts Zhou et al. (2022a); Gal et al. (2022).

In this work, we take interest in the fact that human interaction with these models is often an iterative process. We develop a dataset to study this interaction. The dataset is collected through an interactive game we created where players try to find an optimal prompt for a given task (see Figure 1). In particular, we focus on text-to-image models and ask the player to generate a similar image (*AI*

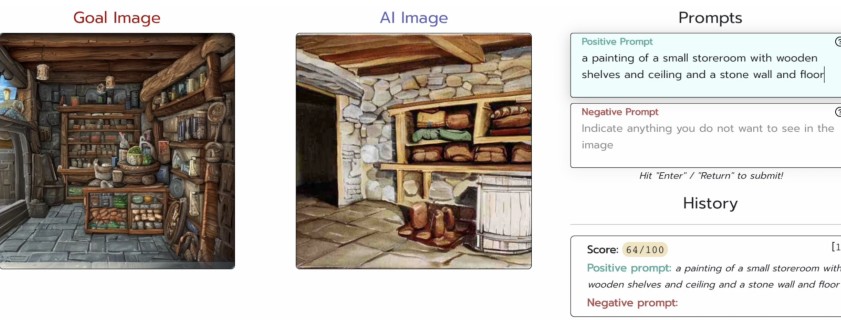

Figure 1: Interface of the ArtWhisperer game. Prompts entered on right. Target (goal) image and player-generated image on left. Previous prompts and scores are displayed in the lower right.

*Image*) to a given *target image*. The player is allowed to iterate on their prompt, using the previously generated image(s) as feedback to help them adjust their prompt. A score is also provided as feedback to help the user calibrate how "close" they are to a similar image.

Using this setup, we collected data on 51,026 interactions from 2,250 players across 191 unique target images. The target images were selected from a diverse set of AI-generated and natural images. We also collected a separate dataset of 4,572 interactions, 140 users, and 51 unique target images in a more controlled setting to assess the robustness of our findings.

Based on this data, we find several interesting patterns in how people interact with AI models. Players discover a diverse set of prompts that all result in images similar to the target. To discover these prompts, players typically make small, iterative updates to their prompts. Each update improves their image with a moderate success rate ($40 - 60\%$ for most target images). Based on these findings, we define and evaluate a metric for model steerability using the stopping time of an empirical Markov model. We use this metric to assess steerability across image categories and across two AI models.

**Ethical considerations** One of the main goals of this work is to help improve the quality of human-AI interaction. Our dataset and findings provide quantitative insights on how people interact with generative AI and can potentially be used to design AI that are easier for people to use. It does not address the broader concern that bad actors may abuse generative AI models.

**Our contributions** We release a public dataset on human interactions with an AI model. To our knowledge, this is the first such dataset showing repeated interactions of people with a text-to-image model to accomplish specified tasks. We also provide an initial analysis of this data and propose a simple-to-calculate metric for assessing model steerability. Our dataset and associated code is made available at [link redacted for submission].

**Related Works** Human-AI interaction datasets for text-to-text and text-to-image models typically focus on single interactions and generally do not provide users with a specific task. Public text-to-image interaction datasets typically contain the generated AI images and/or prompts Santana (2022); Wang et al. (2022) and optionally some form of human preference rating Pressman et al. (2022); Wu et al. (2023); Xu et al. (2023); Kirstain et al. (2023). These datasets generally rely on scraping online repositories like Lexica Lexica or Discord servers focused on AI art. Though some of these datasets include metadata that may allow for reconstruction of prompt iteration, there is no guarantee the user has the same desired output in mind over the iteration. Public text-to-text interaction datasets are much more limited as the best performing models are generally accessible only through APIs with no public user interaction datasets. While some researchers have investigated how human-AI interaction for text-to-text can be improved through various tools Wu et al. (2022a;b), the amount of data collected is limited and not publicly available. There are also repositories containing prompt strategies for various tasks Bach et al. (2022) but no human interaction component.

We seek to rectify two of the shortcomings of the existing datasets–namely, that they do not contain extended interactions as the user attempts to steer the AI, and they do not have a predefined goal. In our work, we create a controlled environment where we allow extended interactions and have a known goal for human users. As shown by our initial analysis, our dataset may enable deeper understanding of user prompting strategies and assessing model steerability.

## 2 INTERACTION GAME

In the game, players are shown a target image. A few example target images are provided in Figures 2, 4. Players are also given a limited interface to a text-to-image model, Stable Diffusion (SD) v2.1 model Rombach et al. (2022b). In particular, players can enter a "positive prompt" (describes the desired content) and a "negative prompt" (describes what should be omitted) to steer the AI model. All models hyperparameters are fixed. Upon inputting a prompt, the player is shown the image generated by the AI model, along with a similarity score between their generated image and the target image. The interface is shown in Figure 1.

### 2.1 HOW TARGET IMAGES ARE SELECTED

We randomly sample target images from two sources. The first is a collection of Wikipedia pages, and the second is a dataset of prompts AI artists have used with SD Santana (2022). In addition to sampling target images, we need to ensure the task is feasible to users. We do not allow users to adjust the seed or other parameters of the model, so we need to ensure the selected model parameters can generate reasonably similar images to the target image. We find that selecting an appropriate random seed is sufficient, and fix all other model parameters (see Appendix A.4 for details and discussion).

**Wikipedia Images** A collection of 35 Wikipedia pages on various topics including art, nature, cities, and various people. A full list of pages sampled from is provided in Appendix 2. From these pages, we scraped 670 figures licensed under the Creative Commons license. These figures were then filtered by which had captions, as well as which images were JPG or PNG images (i.e., not animated, and not PDF files), resulting in 557 images.

For each of the 557 images, we first resize and crop the image to size $512 \times 512$. The Wikipedia caption is used as the ground truth "prompt". Let the image-caption pair be denoted as $(t_i, p_i^*)$. We sample the model on 50 random seeds, with $p_i^*$ as the prompt input. This generates a set of 50 images: $S_i = \{(x_i, s_i) : i = 1, \ldots, 50\}$ for generated image $x_i$ and seed $s_i$. Let $C(x)$ denote the CLIP image embedding Radford et al. (2021) of an image $x$. Then we select the seed as $s_{i^*}$, where

$$i^* := \min_{i=1,\ldots,50} \left\| \frac{C(x_i)}{||C(x_i)||_2} - \frac{C(t_i)}{||C(t_i)||_2} \right\|_2$$

Here, $s_{i^*}$ is selected to minimize the distance to the target image given the target prompt.

**AI-Generated Images** A collection of 2,000 AI-art prompts are randomly sampled from the Stable Diffusion Prompts dataset Santana (2022). For each prompt, $p_i^*$, we generate two sets of images. As before, we use 50 unique random seeds to select the seed, $s_{i^*}$ and an additional 10 random seeds to use for selecting the generated target image (so in total, we use 60 unique random seeds): the first set, $S_{i,1} = \{(x_{i,1}, s_i) : i = 1, \ldots, 10\}$ and $S_{i,2} = \{(x_{i,2}, s_i) : i = 1, \ldots, 50\}$. We select the target image, $t_{i_1^*}$, from $S_{i,1}$:

$$i_1^* := \min_{i=1,\ldots,10} \text{median}\left(\left\{\left\| \frac{C(x_{i,1})}{||C(x_{i,1})||_2} - \frac{C(x_{j,2})}{||C(x_{j,2})||} \right\|_2 : j = 1, \ldots, 50\right\}\right)$$

We select the random seed, $s_{i_2^*}$, using $t_{i_1^*}$ and $S_{i,2}$, with

$$i_2^* := \min_{i=1,\ldots,50} \left\| \frac{C(x_{i,2})}{||C(x_{i,2})||_2} - \frac{C(t_{i_1^*})}{||C(t_{i_1^*})||_2} \right\|_2$$

Here, $t_{i_1^*}$ is chosen to be more representative of the types of images we may expect given the fixed prompt, $p_i^*$. This is because $t_{i_1^*}$ is selected to be close to the center of the sampled images, $S_{i,2}$. The intuition for selecting $s_{i_2^*}$ is the same as selecting $s_{i^*}$ for the Wikipedia images.

### 2.2 SCORING FUNCTION

To provide feedback to players, we created a scoring function to assess the similarity of a player's generated image and the target image. We define the scoring function as

$$score(x, t) = \max(0, \min(100, \alpha \cdot \left\| \frac{C(x)}{||C(x)||_2} - \frac{C(t)}{||C(t)||_2} \right\|_2 + \beta)),$$

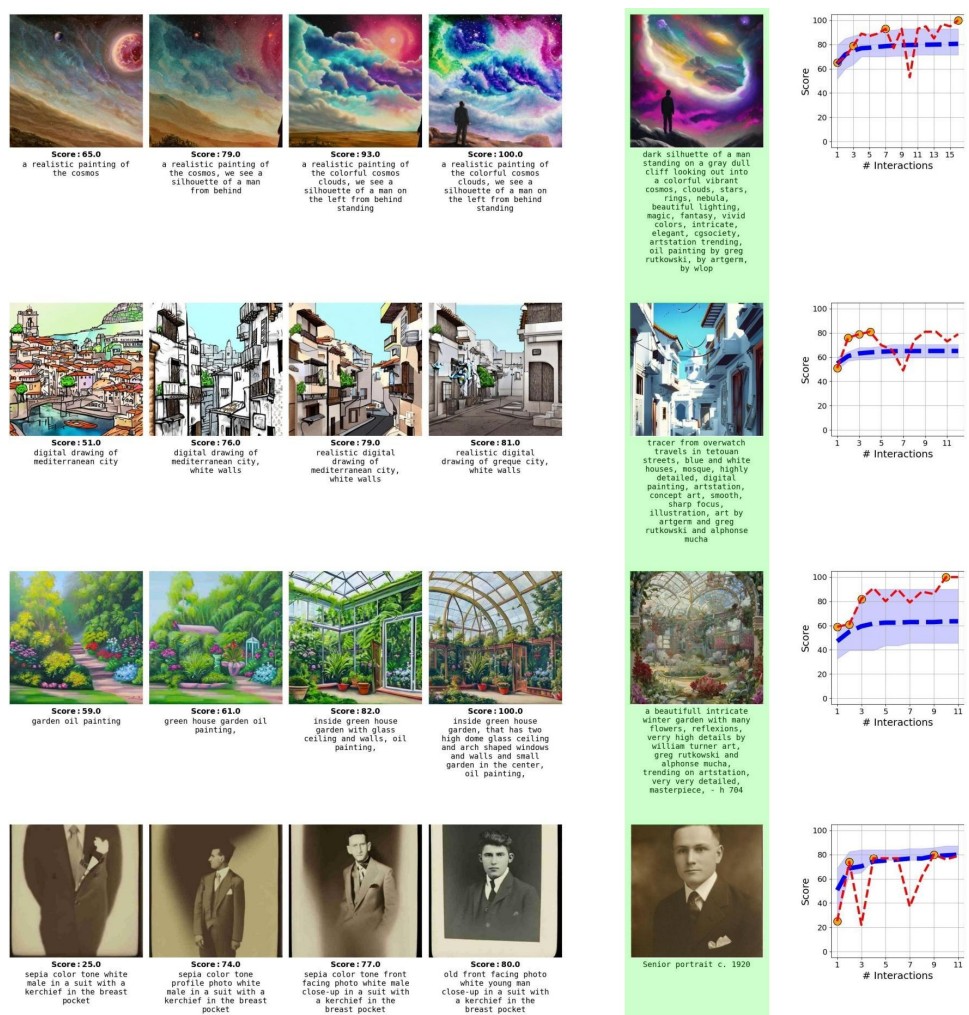

Figure 2: Example user trajectories. In each row, the first 4 images show the prompt progression for a given user. Target image shown in green column. Plot shows the average score trajectory across all users for this target image (blue) and the specific user's full score trajectory (red). Orange circles indicate the displayed images.

for generated image $x$, target image $t$, and constants $\alpha, \beta$. Note the range of $score(x, t)$ is integers in the interval $[0, 100]$. Details on how $\alpha, \beta$ are selected parameters are provided in Appendix A.3.

While this scoring function is often reasonable, it does not always align with the opinions of a human user. To assess how well $score(x, t)$ follows a user's preferences, we acquire ratings from a subset of users (see *ArtWhisperer-Validation* in Section 2.3). We find score$(x, t)$ has a Pearson correlation coefficient of 0.579 indicating reasonable agreement. Further assessment is performed in Section 4.3 and discussed at length in Appendix A.13.

## 2.3 DATASET OVERVIEW

We collected two datasets: *ArtWhisperer* and *ArtWhisperer-Validation*. We use *ArtWhisperer* for most analysis and results; for some of the results in Sections 2.2, 4.2, and 4.3, we also use *ArtWhisperer-Validation* (when referenced). Data was collected from March-May 2023. IRB approval was obtained.

***ArtWhisperer***: A public version of our game was released online, with three new target images released daily. We collected data from consenting users playing the game. Users were not paid. Users were anonymous and we only collected data related to the prompts submitted to ensure privacy of

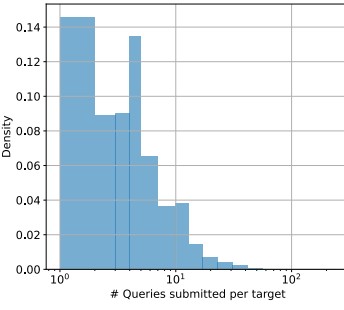 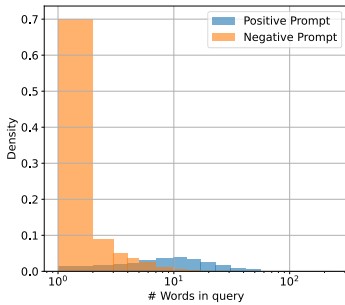

Figure 3: Left, Distribution of # of user queries per target image. The average number of queries per image is 9.18. Right, Distribution of the # of words submitted in a query. The average number of words submitted in a positive and negative prompt are 20.02 and 2.32 respectively.

users. While we expect some users played the game across multiple days, we did not track them. A summary of the *ArtWhisperer* dataset is provided in Table 1. In total, we have 2,250 (potentially non-unique) players corresponding to 51,026 interactions across 191 target images. Players interacted with the model SD v2.1. In Figure 3, we plot the number of queries submitted by players across different target images.

***ArtWhisperer-Validation***: The game (with a near identical interface) was also released as a controlled user study to paid crowd workers on Prolific Prolific Academic. The crowd workers were compensated at a rate of $12.00 per hour for roughly 20 minutes of their time. Workers played the game across 5 randomly selected target images from a pre-selected subset of 51 target images chosen to have diverse content. Workers were also asked to rate each of their images on a scale of 1-10 (i.e., self-scoring their generated images). In total, we collected data on 4,572 interactions, corresponding to 140 users and 51 unique target images across two different diffusion models, SD v2.1 and SD v1.5. Additional details and demographic information are provided in Appendix A.6.

Table 1: *ArtWhisperer* Dataset Overview. Each row contains summary data for a different subset of the dataset. Subsets may overlap. Similar information for *ArtWhisperer-Validation* is in Appendix A.6.

| # Players | # Target Images | # Inter-actions | Average # Prompts | Average Score | Median Duration | Category |
|---|---|---|---|---|---|---|
| **2250** | **191** | **51026** | **9.29** | **58.93** | **18 s** | **Total** |
| 377 | 25 | 3884 | 8.65 | 56.70 | 19 s | Contains famous person? |
| 353 | 32 | 3785 | 8.26 | 61.64 | 21 s | Contains famous landmark? |
| 2005 | 140 | 40290 | 9.24 | 59.83 | 18 s | Contains man-made content? |
| 1177 | 58 | 18255 | 10.93 | 57.21 | 17 s | Contains people? |
| 344 | 77 | 6972 | 8.81 | 62.01 | 20 s | Is real image? |
| 2140 | 103 | 43524 | 9.42 | 58.37 | 17 s | Is AI image? |
| 1483 | 82 | 24913 | 9.14 | 59.45 | 17 s | Is art? |
| 623 | 29 | 7297 | 9.14 | 53.77 | 18 s | Contains nature? |
| 160 | 14 | 1355 | 7.28 | 65.74 | 19 s | Contains city? |
| 1239 | 39 | 15872 | 9.91 | 56.74 | 16 s | Is fantasy? |
| 618 | 19 | 8359 | 10.51 | 57.88 | 17 s | Is sci-fi or space? |

# 3 PROMPT DIVERSITY

We quantify prompt diversity by looking at the distribution of prompts in the text embedding space. In particular, we use the CLIP text embedding Radford et al. (2021), though we do find the choice of embedding is not particularly important for our results (see Appendix A.7).

### 3.1 DIVERSE PROMPTS USED FOR HIGH SCORES

People achieve high scores with a diverse set of prompts. It is not surprising that this is possible (i.e., that the score metric has multiple local maxima), but it is potentially surprising that users consistently find these differing local maxima. Examples are shown in the four leftmost columns of Figure 4.

We quantify this finding in Figure 5, where we plot two metrics defined as follows. Let $z_0, z_n$ be normalized embeddings of the initial and best prompt/image found by a user. Let $z^*$ be the normalized embedding of the target prompt/image. We define the difference in embedding distance to ground truth as $||z_n - z^*||_2 - ||z_0 - z^*||_2$. In blue, we use the CLIP text embeddings of the prompts; in orange, we use the CLIP image embeddings of the generated images. We note two findings here: (1) the metric applied to the image embeddings is guaranteed to be non-positive as the embedding distance is monotonically decreasing with the score, and (2) the metric applied to the text embeddings is apparently symmetric around 0, indicating that unlike the image embedding, distance in the text embedding space does not monotonically decrease with score. Together, these findings illustrate that users tend to discover diverse prompts and *do not converge in their prompt design*.

The right two columns of Figure 4 provide another visualization at an individual level. Here, we plot a UMAP McInnes et al. (2018) projection for the CLIP image and text embeddings of the target image and a sample of the submitted prompts (and corresponding generated images).The target embedding is in orange, the first prompt embedding is in red, and the last (best scoring) prompt embedding is in green. The arrows connect a given user's first and last prompt. We see that a decrease in distance in the image embedding space (which is inversely correlated with the user's score) does not always correspond to a decrease distance in the text embedding space.

Additionally, we find the distribution of prompts does not converge. In the left of Figure 6, we plot the distribution of distances between the first prompt (in blue) and the last prompt (in orange) to the average prompt for the corresponding target image. Despite the average score improving from 51.9 to 70.3 (out of 100) indicating a significant improvement in score, prompt diversity does not significantly diminish. That is, users do not converge to similar prompts to achieve high scores. Similar analysis of the image embedding space suggests image diversity *decreases* (Figure 5).

### 3.2 PEOPLE SUBMIT SIMILAR PROMPTS THROUGHOUT THEIR INTERACTION

In the center of Figure 6, we plot the distribution of the standard deviation of prompts for users (blue) and for permuted users (orange). Permuted users are generated by sampling from all prompts for a given target image uniformly, using the same distribution of number of prompts as for real users. The gap between the two distributions shows that individuals do not randomly sample prompts each interaction, but base new prompts off of previously submitted prompts (p-value $< 10^{-10}$, t-test for independent variables). An analysis of how scores change between adjacent prompts shows that this strategy has a moderate success rate and improves the score $40 - 60\%$ of the time, with an average rate of $48.6\%$ (note that score changes $< 1$ are counted as unchanged; this occurs $10.2\%$ of the time).

While this is not a surprising result (that users often do not make significant changes to their prompt), but it is an important result to understand how typical users interact with AI models. Moreover, it suggests that user initialization (i.e., the first prompt they submit) is critical.

### 3.3 PEOPLE HAVE SIMILAR PROMPT STYLES ACROSS IMAGES

We quantify user style by computing the difference (in the CLIP text embedding space) between the average prompt of a given user and the average prompt across all users for a given target image. To quantify style variation for a user, we then compute the standard deviation of the user style across the target images the user generated. In the right of Figure 6, we plot the distribution of user style variation for real users (blue) and permuted users (orange). Permuted users are generated by randomly sampling user styles. This allows us to test whether users have a consistent prompting style. We find users do indeed have specific styles of prompting (p-value $< 10^{-10}$, t-test for independent variables). However, the difference is not seemingly not large, suggesting that while user style may a component to prompting, other factors related to the target image may be more important.

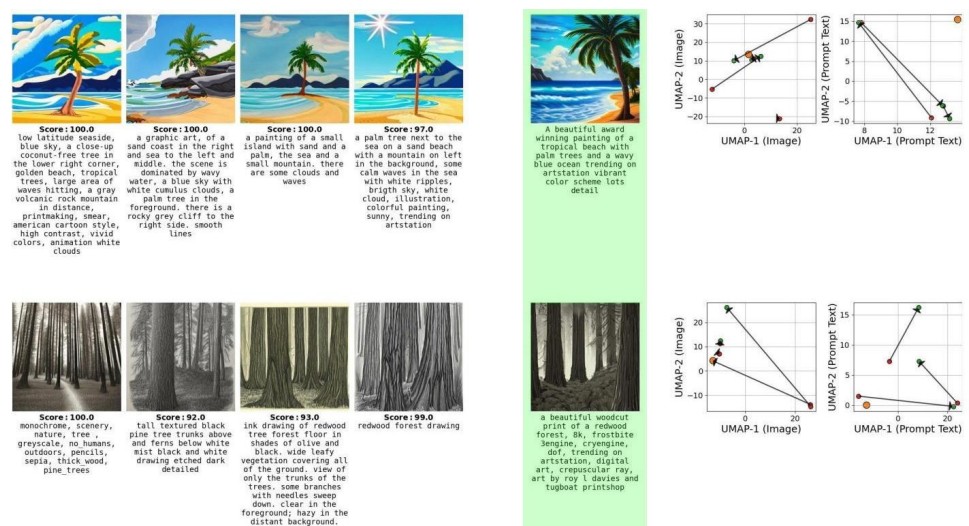

Figure 4: Left: Diverse, high-scoring prompt submissions from different player. Target image in green column. Right: Image (left) and text (right) embeddings of displayed images (target in orange; user submissions in green (best) and red (first)) using the UMAP McInnes et al. (2018) projection.

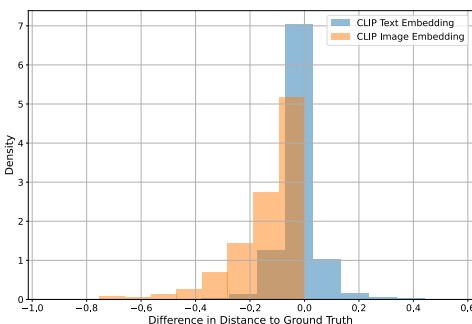

Figure 5: Difference of distance from the first prompt to ground truth and distance from the last (best) prompt to ground truth for CLIP text (blue) and CLIP image embeddings (orange).

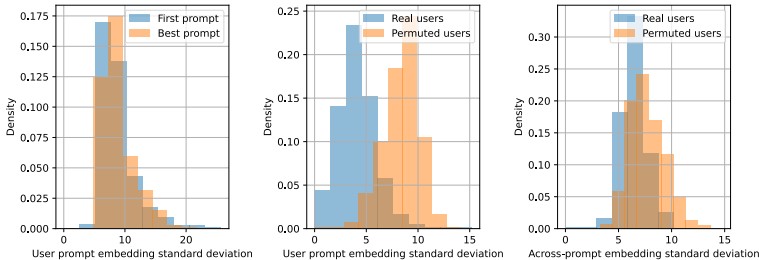

Figure 6: Left: Users submit diverse (across users) prompts, both at beginning and end of interaction. Center: Individual users do not submit diverse prompts. Right: Users have different prompting styles.

## 4 MODEL STEERABILITY

Model steerability refers to the ability of a user to steer a model towards a desired outcome. There is no current consensus on how to measure AI steerability. A common approach is to simply measure performance of a model on standardized dataset evaluations Jahanian et al. (2019); OpenAI (2023). While this can enable comparisons between tasks and models, this approach does not allow for the feedback loop present when humans interact with a model. Steerability can also be measured qualitatively based on user assessment of their experience interacting with the AI Chung et al. (2022). We create a simple yet informative measure of model steerability. We then analyze this measure across different subgroups of images and across two different Stable Diffusion models–SDv2.1 and the older SDv1.5 Rombach et al. (2022a).

### 4.1 MEASURING STEERABILITY

As discussed in Section 3.2, users typically engage with the model through clusters of similar prompts. They typically start with an initial base prompt and proceed to make multiple incremental modifications to it. We use this observation as a basis for creating a steerability metric. We define a Markov chain between scores. Each node is a score with edges connecting to the subsequent score. To make this tractable for empirical analysis, we bin scores into five groups: $[0, 20], [21, 40], [41, 60], [61, 80], [81, 100]$. We use the expected time taken to reach the last score bin, $[81, 100]$, as our steerability score (i.e., the stopping time to reach an adequate score).

For each target image, we calculate the empirical transition probability matrix between binned scores using all the players' data for that image. We then calculate the *steerability score* for the given target image by running a Monte Carlo simulation to estimate stopping time, as defined above. To assess steerability across a group of images, we average steerability score across all images in the group.

### 4.2 ANALYSIS

In Figure 7, we plot the steerability score across image groups. Error bars show the standard error. For examples of steerability scores for individual images, see Appendix A.11. We find that images containing famous people or landmarks, real images (not AI generated), contain cities, or contain nature are the most steerable. AI-generated images, fantasy images, and images of human art are the least steerable. There are a few possible explanations. The model we are assessing here, SDv2.1, as well as its text encoder OpenCLIP, are trained on subsets of LAION5B Schuhmann et al. (2022). The contents of LAION5B are predominantly real world images, indicating why these images may be more steerable (i.e., text describing these types of images may have a better encoding). Moreover, the prompts for AI-generated images and fantasy images generally include specific internet artists and/or art styles which may not be known to most users making achieving the desired target image more difficult. Another potential reason is the distribution of images chosen for each category. Clearly, there are "easier" and "more difficult" images in each category; part of the reason for smaller stopping time may be the sample of images chosen rather than the actual image category.

Using the *ArtWhisperer-Validation* data, we also compare steerability across two models: SDv2.1 and SDv1.5. Across most image categories, we observe a similar steerability. Images of nature, sci-fi or space, and real images have the largest differences in steerability between the two models; SDv2.1 is more steerable in all three cases. This suggests that SDv2.1 may be more steerable for natural images as well as sci-fi images, and is similarly steerable for other kinds of images including AI-generated artwork. One explanation may be that most of our users were not aware of certain prompting strategies that help models generate more aesthetic images or certain art styles; it is possible that for experienced users, AI art images may be more steerable, and differences between models may be magnified if, for example, a user is experienced working with one particular model. More discussion is provided in Appendix A.10.

### 4.3 JUSTIFICATION FOR AUTOMATED SCORE

One limitation of our steerability metric comes from the method of scoring user-submitted prompts. Ideally, we would assess steerability with a user's personal preferences. As mentioned in Section 2.2, the scores and human ratings have a positive correlation. Here, we use the human ratings from

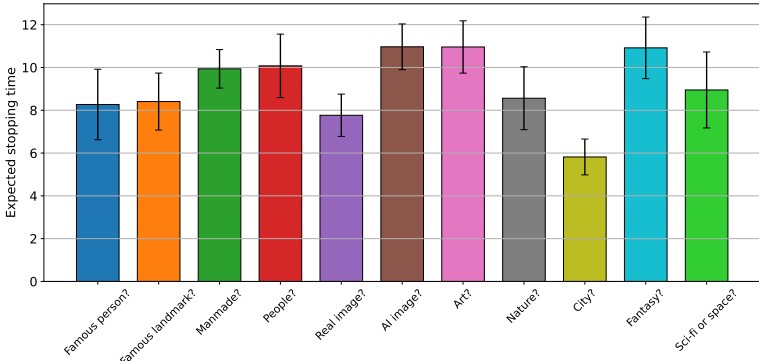

Figure 7: Steerability across image groups (smaller indicates increased steerability). Bars show average expected stopping time across images in the image group; error bars show standard error.

*ArtWhisperer-Validation* instead of our score function to assess steerability. We compute the steerability score across both models and across image groups. Generally, the steerability scores change little. In all but two cases (SDv2.1 on sci-fi and space images; SDv1.5 on nature images), the human rating-based steerability score remains within a 95% confidence interval of the score-based steerability score. While our score function may not perfectly capture human preferences, the steerability score we generate appears to be robust to these issues. Further discussion is included in Appendix A.13.

## 5 DISCUSSION

As demonstrated in our analysis, the *ArtWhisperer* and *ArtWhisperer-Validation* datasets can provide insights into user prompting strategies and enables us to assess model steerability for individual tasks and groups of tasks. What makes our dataset particularly useful is the controlled interactive environment, where users work toward a fixed goal, that we capture data in.

One of the most exciting use cases we see for our dataset is to create synthetic humans for prompt generation. For example, similar to the method described in Promptist Hao et al. (2022), we imagine fine-tuning a large language model with our dataset to generate prompt trajectories (i.e., rather than an optimized prompt) using similar exploration strategies as a human prompter. These synthetic prompters could be based on multimodal models like OpenFlamingo Awadalla et al. (2023) or text-only models and use score-feedback to condition the trajectory generation. As an initial proof-of-concept, we fine-tuned a MT0-large model Muennighoff et al. (2022) model on our dataset and found the fine-tuned model can indeed behave similarly to human users (see Appendix A.14). These synthetic prompters have several potential use cases:

1. Automating measurement of text-to-image model steerability by using synthetic users in place of real human prompters. While we believe our proposed steerability metric is effective, its main limitation currently is the requirement for human annotations.

2. Incorporating steerability in the objective function for text-to-image models. By representing steerability as a function of synthetic users, it becomes possible to explicitly optimize a model for steerability.

3. Generating human readable image captions that are compatible with a Stable Diffusion model by using the synthetic prompter to optimize the token representation of the prompt.

Additionally, our dataset can be used for further analysis on human prompting strategies beyond what we discussed in the paper. For example, one question we only touched upon is whether we can compare human prompters to automated prompt optimization methods (e.g., do humans behave similar to some gradient-based optimization approach in the prompt embedding space?). There are also potential uses for crafting better image similarity metrics using the human ratings we collected.

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
