# OpenReview forum: "ArtWhisperer: A Dataset for Characterizing Human-AI Interactions in Artistic Creations"
_ICLR.cc/2024/Conference — Submitted to ICLR 2024_

### Official Review · Reviewer_kk5o · 2023-10-29

**Soundness:** 4 excellent
**Presentation:** 4 excellent
**Contribution:** 3 good
**Rating:** 6
**Confidence:** 4

**Summary:**

This paper aims to study how people write prompts for generating a target image with text-to-image models. The authors have collected the ArtWhisperer dataset that includes the different iterations of prompts a user tries out in generating the images, and from this dataset they have observed that different, diverse prompts can be used to generate an image. They also propose a metric for quantifying the steerability of the target image generation task.

**Strengths:**

* The paper addresses an interesting problem that has not been really investigated by other researchers in depth.
* The dataset will be made available to the public. To the best of my knowledge, no comparable dataset exists.
* The authors have followed responsible research practices (getting IRB approval, handling user data/privacy properly).
* The work is well written and easy to follow.
* The discussed future use cases including synthetic prompting and fully automated evaluation of t2i models would be relevant to many real-world applications.

**Weaknesses:**

* The work does not investigate the behavior of users with malevolent intent (the authors acknowledge this), as their actions may not align with those of good actors. The users are assumed to be the latter (or at least, homogeneous in that respect), and the observations made in this paper are based upon that assumption.
* I would be interested to see more discussion/applications of how to take these noted observations of user behavior, steerability, and prompts and apply them to an existing problem. These insights could have possible implications for model training/evaluation in the future.
* There may have been returning users on a day-to-day basis, so it’s feasible that a user became more experienced with prompting over time, which would thus affect steerability. The authors however did not collect this type of data, so it’s not possible to observe this.

**Questions:**

1. Do the authors have any sense for how user behavior may differ with other t2i models? For example, does the quality of the model (i.e., its ability to generate high-quality imagery) affect the results?
2. Did the authors consider enabling users to adjust the model parameters (e.g. the seed)? This could potentially enable the user to generate an image that is even closer to the target, and could also be useful in future applications highlighted in the Discussion section.

---

> ### Author Response · Authors · 2023-11-17
> **Thank you for your helpful feedback!**
>
> Questions answered:
> - As mentioned in the discussion and appendix (A.14), we have demonstrated that our dataset can be used to train synthetic users by fine-tuning an LLM (we used an MT0 model here). The validation we did suggests the fine-tuned model can behave similarly to human users with respect to score trajectory / steerability. While we leave much of this to future work, it is easily conceivable that synthetic users trained on user interaction data (e.g., our dataset) can be used to fine-tune VLM models like stable diffusion to increase steerability (e.g., using our metric to assess steerability).
>   - We have included more discussion on this point in the revision.
> - As you point out, we were unfortunately unable to collect long term data on repeated user interactions. In the controlled user study, we did collect interactions across a number of tasks. From this data, we did not find a significant increase in steerability (after correcting for steerability differences across images). While intuitively we do believe steerability should increase over time, our dataset suggests that this increase is more subtle. We suspect additional feedback given to the user (e.g., instructions to improve / correct their submitted prompts) may more significantly increase steerability, but we did not test this hypothesis here.
> - Our intuition is that model quality is a significant factor in steerability. In particular, higher quality image output increases steerability (up to a point, as image quality is not the only factor in steerability).
>   - This is easily seen in the following extreme case – a user who does not look at the model output (i.e., the model output quality is low) can still steer the model through random changes to the prompt – whether the change improves or decreases the score is seemingly random, however, limiting steerability (this assumes that the model encodes the user prompt in the image, despite its low visual quality; if this does not happen, the scoring will be even more random).
> - We deliberately decided to limit user input to just the prompt. This was done for a few reasons (added this additional discussion to Section 2 and the Appendix A.4 in the revision):
>     1. We wanted all users who enter the same prompt for a given image to see the same output.
>     2. We wanted to limit the complexity of the task for users less/unfamiliar with text-to-image models.
>     3. We wanted users to generate new prompts, not simply resample new seeds until getting lucky. (While users could still employ a version of this “random resampling” strategy by making small changes to their prompts, we did not want to encourage it through a seed parameter. This random resampling strategy, while useful in practice, is not such an interesting result for research purposes as it is easy to simulate random resampling strategies without any user input.)

---

### Official Review · Reviewer_Ww77 · 2023-10-29

**Soundness:** 3 good
**Presentation:** 4 excellent
**Contribution:** 3 good
**Rating:** 5
**Confidence:** 4

**Summary:**

This dataset studies how people interact with text-to-image models to generate desired target images. Particularly, during the data collection, the job of human is to iteratively re-fine the text prompt (only) to create a similar-looking image (to a given reference). Such a procedure traces the trajectory of human in making creative art using text-to-image model, and presents a unique sequential dataset for studying human-AI collaboration.

Then this paper perform analysis on the collected dataset, and find that user is able to discover a variety of text description that generates similar images, with the trajectory of user queries of similar diversity along the search. Then they propose a metric to quantify the steerability of AI, which is the expected number of interaction required to adequately complete a task.

**Strengths:**

- The paper is well presented, the motivation, data collection, experiments are clearly written. There is abundant details to understand the quality of the data, the distribution of user trajectory, the domain of data, etc.

**Weaknesses:**

- It's unclear to me on how we could utilize the study done in this paper to improve research in text-to-image generation modeling, or Human AI interaction research. I saw some direction from the conclusion but those can be drawn without studying the research presented in this paper.

**Questions:**

- In the section about AI-generated images, it seems like from this seed selection procedure, you are only choosing the target image based on how easy it could be re-created using the prompt (and different random seeds). The criteria is not encouraging the aesthetic of the image, nor alignment between image and target prompts?
- What is IRB approval?

---

> ### Author Response · Authors · 2023-11-17
> **Thank you for your helpful feedback!**
>
> There are multiple uses for our dataset and data collection procedure as discussed throughout the paper and discussion sections:
> 1. Better understanding of user prompting strategies and diversity of user prompts
>     - To our knowledge, this is the first public dataset capturing how users iterate on their prompt designs for text to image models. This dataset lets us study how people perform this iteration and how we might improve models to accommodate human usage (e.g., see Sections 2, 3 and Figures 2,4,6).
>         * For example, we find using our dataset that users arrive at diverse prompts for the same image – they do not converge to the same image description (Section 3 and Figure 6).
>         * As another example, we also find evidence using our dataset that there is a distinct style each user has (though the effect of style on steerability may depend more on specific image) (Section 3 and Figure 6).
> 2. Quantifying model alignment to users in a rigorous way through a steerability metric.
>     - We use our dataset as the basis for defining a steerability metric, and propose our data collection procedure (or an analog with simulated users) can be used to assess models that require human interaction. (Section 4, Figure 7)
> 3. Synthetic user agents
>     - As we describe in the discussion section, we believe an important use case of our dataset is to create synthetic users. We test and validate this approach where we fine-tune an MT0-large model to replicate a human prompter (using numbering from the revision: Appendix A.14, Figures 21,22,23). Creating synthetic users that *employ real human prompting strategies / vocabulary* has a clear benefit to model training: we can incorporate this into loss metrics to increase alignment between models and human users across an interaction.
>
> Questions answered:
> - The seed selection (and target image selection) procedure is used to find (1) a target image and (2) a user seed such that user prompting variation does not significantly alter the user generated image similarity to the target. (Minimal) additional (subjective) quality control is performed by the authors to ensure image content is reasonably aligned with the prompt.
> - IRB approval is a standard process for ensuring ethical research when involving human subjects.
>
> Thank you again for your time! We hope our responses have addressed your questions and that you would consider raising your score. Please let us know if you have additional questions and we are happy to respond further.

---

### Official Review · Reviewer_oZkf · 2023-11-01

**Soundness:** 3 good
**Presentation:** 2 fair
**Contribution:** 3 good
**Rating:** 5
**Confidence:** 3

**Summary:**

The authors introduce a dataset for understanding prompt tailoring in image generation. To collect the data, they introduced a game collecting 51k interactions across 2k papers, with a smaller group of paid users. The game is driven by two datasources AI Pompts and Wikipedia which they align using CLIP embeddings to select optimal plausible query images. From the game, they identify Markov chain approach to the steerability of user prompts. The authors envisage the dataset can be used to improve the outputs of image generation methods but also as a synthetic test setting using tailored NLP methods to test the response.

**Strengths:**

- Prompt engineering is a difficult problem to capture which the authors have proposed a method to achieve future analytical comparisons of methods.
- The dataset seems to be sufficient to capture user insights as the authors identify steerability is achievable.
- The use of multiple data sources to drive the method is good, especially as real world image captions do not contain the content description. However, the authors fail to quantify the performance, and they do not elaborate in the evaluation of the two sources.

**Weaknesses:**

- It would be better to be clearer about the controlled collection earlier, simply paid and unpaid would avoid assumptions of how this was performed.
- The authors are missing a comparison of how true the distribution of generated images is to the original images as the Wikipedia captions in general don't describe the content so highly likely they do not match the original image. A quantitative evaluation of this is important otherwise it makes the choice of Wikipedia not relevant and does not increase validity over using any source.
- Authors should avoid re-using mathematical notation x_i changes through section 2. Although in general x is re-used in changing context throughout the paper.
- Lack of discussion on participant characteristics how was gender, race & ethnicity balanced?
- Steerability contribution is well received, however, this would have more validity if was tested on a different model to confirm the insights, two versions of StableDiffusion provide limited insight.,

**Questions:**

- How was bias addressed during data collection?
- How do the results on steerability differ between the two input datasets?
- Does the influence of unpaid/paid effect the outcomes?

**Details Of Ethics Concerns:**

The potential dataset has the risk of large bias. The authors should justify how bias is addressed in the data collection to avoid future issues around this topic being introduced. The minimum should be clearly stating the diversity of participants in data collection. However, if this reveals large imbalance the data needs to be re-balanced or studied to understand if the concluding steerability approach is bias.

---

> ### Author Response · Authors · 2023-11-17
> **Thank you for your helpful feedback!**
>
> Questions answered:
> - We have clarified the two datasets better in the revision (see Section 2.3).
> - Comparison of the generated images from the target prompt with the real Wikipedia images is added in the Appendix. We also added a sentence clarifying that the metric we use is calibrated such that using the Wikipedia caption gives a perfect score for the Wikipedia image (this calibration is also done for generated AI images).
> - The reason for using Wikipedia images is to collect and compare the steerability of a broad variety of images – both AI art as well as real art (e.g., paintings or sculptures) and other natural images (cities, wildlife, etc.).
> - Notation has been clarified throughout the paper. In Section 2.1, $x_i$ is used for Wikipedia images and $x_{i,1}$ and $x_{i,2}$ are used for AI-generated images. In Section 2.2, we refer to images using just $x$. In Section 3, we have changed notation to use $z$ instead of $x$ to describe image/text embeddings.
> - Participant demographics were intentionally not collected to avoid identifying users.
>     - In the controlled user study, demographics were collected and a supplemental figure has been added. Users were sampled through Prolific to a fixed 50/50 split of male and female users. In the revision, we have included histograms of the age, sex, and ethnicity distributions in Figure 11 in the Appendix (A.6). We did not find significant differences in steerability across these demographics.
> - While comparison with additional models would be nice, we believe our results are still a valuable contribution to the community.
>     - The two stable diffusion models we use are in fact quite different and produce differing results given the same prompt (note, for example, we needed to re-select the user seed for the second model to ensure the model produced a similar enough image when using the target prompt; using the same seed as the original model could result in vastly different images). So, we believe the comparison we provide is interesting in itself.
>     - Moreover, it also provides evidence of differences in steerability between two models and that we can measure these differences using our data collection method and proposed steerability metric.
> - Steerability results between Wikipedia and AI art datasets are compared in Figure 7. Note the differences between the “AI Art” (AI Art dataset) and “Real Images” (Wikipedia dataset) indicate that real images are more steerable.
> - We did not find any significant differences between paid and unpaid users.
>
> Thank you again for your time! We hope our responses have addressed your questions and that you would consider raising your score. Please let us know if you have additional questions and we are happy to respond further.

---

### Official Review · Reviewer_1NDi · 2023-11-05

**Soundness:** 2 fair
**Presentation:** 3 good
**Contribution:** 2 fair
**Rating:** 5
**Confidence:** 4

**Summary:**

The paper investigates human-AI interactions in the context of text to image generation models. The paper presents a gamified experimental setup called “ArtWhisperer”, where a subject is provided with a target image. The subject’s goal is to iteratively prompt the model to successfully generate an image similar to the given target image.

The paper presents analyses of the experimental data, and defines a metric called “steerability”, which captures the human subject’s difficulty of generating an image via prompting.

**Strengths:**

1. The experimental data of goal-driven prompt-interactions appears to be unique, and of potential interest to the generative AI community.

2. The overall experimental setup seems reasonable — the generated images are evaluated against *generated* images; not real images. The controlled setting, is especially good, and provides important additional insights about the overall task.

3. The paper clearly discusses the major limitation of the experimental setup — relatively low (although positive) correlation between human notions of similarity and the automatic score. This is first identified via a different experimental setup, and the consequence of the mismatch is quantified via a steerability score on a smaller set of images based on human-provided similarity scores. The difference in steerability scores on this smaller subset is not too large. This is promising, and improves confidence about the overall validity of the conclusions based on experiments with automatic scores.

4. The paper analyses the data to describe findings that may be of interest to researchers and practitioners working on human-AI interaction applications. Specifically, the study of steerability is quite relevant for creative AI applications.

**Weaknesses:**

1. Prompt engineering is now a topic of interest for people who wish to efficiently generate usable content from conditional generative models. My hypothesis is that, “Steerability”, i.e., the ease of generating the target image, depends on the amount of experience of the user. It would be great if the authors discuss how they might be able to decouple the influence of returning users who potentially have the ability to efficiently “steer” the generative model via better prompts.

2. Apart from the prompting experience, a person’s knowledge of the contents of the target image could also play a role in steerability. Especially in cases of a famous person, famous landmark, famous city, etc. There appear to be some person-specific confounding factors that affect steerability, which the paper doesn’t seem to address. It appears that the rationale for this is that, with a large enough population of users, such person-specific effects might be negligible. However, it would be nice if these confounding factors were explicitly identified and discussed.

3. “Image specificity” (Jas et al., CVPR 2014) measures the extent to which human captions that describe an image, varies across people. The observation that target images containing specific landmarks are more steerable than abstract images, appears to allude to a similar concept. Of course, the idea of steerability involves additional complexity of the generative model, and the human’s understanding of prompting. However, it seems possible that images with high “specificity” might lend themselves to be more steerable. It would be great to discuss this further.

4. The mismatch between human notions of similarity and the automatic metric makes the experimental setup suboptimal, and data a little more complicated to draw conclusions from. Please see “Questions” for specific concerns regarding the mismatch.

References:
(Jas et al., CVPR 2014) Jas, Mainak, and Devi Parikh. "Image specificity." *Proceedings of the IEEE Conference on Computer Vision and Pattern Recognition*. 2015

**Questions:**

1. Were there comments / feedback from users that elicited insight regarding their strategy? Specifically regarding the incongruity between human similarity notions and automatically computed similarity score? E.g., did certain persons try to maximise the automatic score while others maximised their own perception of similarity?

2. Was there an incentive for high score? E.g., additional reward? Do we know that people were taking the task “seriously”, and not just experimenting / exploring?

3. What is the score distribution between each generated image (and the target image) — that are generated from the same prompt? Is there any insight into the variance in matching score across different random seeds of the model (keeping the prompt constant)

4. “The intuition here is that t*1_i is more representative of the types of images we may expect given the fixed prompt, p*_i .” — the intuition here is unfortunately not clear to this reader. Could the authors please elaborate? Thanks!

---

> ### Author Response · Authors · 2023-11-17
> **Thank you for your helpful feedback!**
>
> Influence of returning users on steerability
>   - In the controlled user study, we did not observe a significant change in the model’s steerability (controlling for target image) over time. This suggests that user experience did not have a large impact on steerability.
>   - Our work provides insight into only the immediate steerability of a model, and provides evidence that absent specific feedback (e.g., instructions on how to steer a model), people do not improve steerability very quickly (within 5 examples). It might be possible that a much longer experience can improve steerability.
>
> Knowledge of an image subject improves steerability
>   - This is partially due to the prompt supplied by the user (the user can provide a more specific prompt) and partially due to limitations of our image similarity metric (in some cases, the metric can be weighted to the subject matter, and referencing the correct subject in the prompt can lead to a higher steerability score). A figure (Figure 18 in the Appendix) was added to the revision. It shows a significant difference when the user is familiar with the subject of the target image (defined as the user referencing the key subject in at least one of the prompts they submit).
>   - People were instructed to and did attempt to maximize the synthetic score. We did not collect separate comments from users.
>   - As discussed in Appendix A.4, users in the controlled user study were given a bonus for achieving higher scores.
>       - Users generally took the task seriously. A prefiltering step was used that removed data where a user did not achieve beyond a certain minimum score (these were identified as cases where a user was simply exploring and not trying to complete the task)
>   - As variance of image output (for a given target prompt) increases, so does the steerability score (larger steerability score corresponds to more difficulty in steering the model). In other words, if repeatedly sampling a model with the same prompt can produce a wide range of outputs (high variance), then the model is likely less steerable for the content in those generated images. We added a figure (Figure 17 in the Appendix) and a brief discussion on this point in the Appendix of the revision.
>   - The variance largely depends on the image – for some images it is quite large while for others it can be quite small. This is the reason for our procedure to select the target image to show users and the fixed seed we provide for users. The target image and seed are selected to minimize the difference between the target image and other images generated using different seeds with the same target prompt.
>       - The intuition here (why $t_{i_1^*}$ is more representative) is the following. $t_{i_1^*}$ is the target image closest to the center (using median) of the set of candidate target images (using a fixed target prompt). While the model may produce a wide range of images for the given target prompt, we pick the target image that is the “average” one (using median instead of mean to reduce the effect of outliers). By picking this “average” image, we also hope to reduce the difference of images generated using similar prompts. We have clarified this in the revision.
>
> Thank you again for your time! We hope our responses have addressed your questions and that you would consider raising your score. Please let us know if you have additional questions and we are happy to respond further.

---

### Meta-Review · Area_Chair_KZ71 · 2023-12-14

**Metareview:**

The paper presents a comprehensive study of human interactions with AI in the context of artistic creation. The authors introduce a novel dataset, ArtWhisperer, built from over 50,000 human-AI interactions. This dataset captures the iterative process users undergo to find prompts that lead AI to generate images similar to a given target. The paper proposes a metric for steerability, defined as the expected number of interactions required to complete a task, and compares steerability across different image types and AI models.

Reviewer 1NDi acknowledges the uniqueness and potential interest of the dataset for the generative AI community but raises concerns about the influence of user experience on steerability and the lack of discussion on person-specific factors affecting steerability. Additionally, they points out the need for more discussion on the concept of "image specificity" and its relation to steerability. In response, the authors clarified the methodology and added supplementary data to support their findings. Reviewer oZkf appreciates the approach to capturing prompt engineering and the use of multiple data sources. However, oZkf criticizes the lack of clarity on controlled data collection, the absence of participant demographic details, and the limited comparative analysis with different models. In their rebuttal, the authors provided further clarifications on dataset comparisons, demographic details in the user study, and defended their choice of models, highlighting the differences between the two SD models used. Reviewer Ww77 questions the paper’s utility in advancing text-to-image generation modeling. The authors responded by highlighting multiple uses of their dataset and data collection procedure, including better understanding of user prompting strategies, quantifying model alignment to users, and creating synthetic user agents. Reviewer kk5o suggests more discussion on the application of observations to existing problems and the behavior of users with malevolent intent. The authors addressed these points by discussing potential applications and limitations of their dataset. After the rebuttal, kk5o revised their rating to a borderline accept, but noting the clear need for additional discussion and clarification.

Overall, while the paper presents good contributions to the field of Human-AI interaction, particularly in artistic contexts. However the reviewing process also shows several aspects that warrant further exploration, discussion and analysis. After a thorough review of the manuscript, the reviewers' feedback, and the authors' rebuttal, the AC recommends rejection at this time. The authors are encouraged to incorporate reviewers' comprehensive and constructive feedback and do another round of substantial revision, for a future submission.

**Justification For Why Not Higher Score:**

The decision to reject stems from the fact that the paper really requires a major revision to address all reviewer concerns in a more substantial way, particularly around the practical applications and implications of the study. The difference between pre- and post-rebuttal assessments shows a positive shift in some reviewers' perspectives, but not sufficiently to overcome the initial concerns at this time.

**Justification For Why Not Lower Score:**

N/A

---

### Decision · Program_Chairs · 2024-01-16

Reject